# Importance of ABC Transporters in the Survival of Parasitic Nematodes and the Prospect for the Development of Novel Control Strategies

**DOI:** 10.3390/pathogens12060755

**Published:** 2023-05-24

**Authors:** Ali Raza, Andrew R. Williams, Muhammad Mustafa Abeer

**Affiliations:** 1Queensland Alliance for Agriculture & Food Innovation, Centre for Animal Science, The University of Queensland, St Lucia, QLD 4067, Australia; 2Department of Veterinary and Animal Sciences, Faculty of Health and Medical Sciences, University of Copenhagen, 2200 Frederiksberg, Denmark; 3CSIRO Food Innovation Centre, Werribee, VIC 3030, Australia

**Keywords:** ABC transporters, P-glycoproteins, parasitic nematodes, physiology, anthelmintic resistance, multidrug resistance inhibitors

## Abstract

ABC transporters, a family of ATP-dependent transmembrane proteins, are responsible for the active transport of a wide range of molecules across cell membranes, including drugs, toxins, and nutrients. Nematodes possess a great diversity of ABC transporters; however, only P-glycoproteins have been well-characterized compared to other classes. The ABC transport proteins have been implicated in developing resistance to various classes of anthelmintic drugs in parasitic nematodes; their role in plant and human parasitic nematodes still needs further investigation. Therefore, ABC transport proteins offer a potential opportunity to develop nematode control strategies. Multidrug resistance inhibitors are becoming more attractive for controlling nematodes due to their potential to increase drug efficacy in two ways: (i) by limiting drug efflux from nematodes, thereby increasing the amount of drug that reaches its target site, and (ii) by reducing drug excretion by host animals, thereby enhancing drug bioavailability. This article reviews the role of ABC transporters in the survival of parasitic nematodes, including the genes involved, their regulation and physiological roles, as well as recent developments in their characterization. It also discusses the association of ABC transporters with anthelmintic resistance and the possibility of targeting them with next-generation inhibitors or nutraceuticals (e.g., polyphenols) to control parasitic infections.

## 1. Introduction

ATP binding cassette (ABC) transport systems, one of the largest protein superfamilies, are found in all three domains of life (prokaryotes (archaea and bacteria) and eukaryotes). Eukaryotic ABC transporters typically consist of two conserved domains, a transmembrane domain (TMD) and a nucleotide-binding domain (NBD) [1]. Based on genomic analysis, eukaryotic ABC transporters have been categorized into seven sub-families from A to G [2,3,4]. Two of the ABC transport protein families (ABCE and ABCF) lack TMDs, and thus do not act as transporters, and are associated with other cellular processes, for example, ribonuclease inhibition and translational control [5,6]. ABC transporters have diverse structures depending on different domain compositions and ATP binding sites; thus, ABC transporters are classified as full transporters (complete structure with two NBDs and two TMDs) containing two ATP-binding sites and the half transporters (half structure with one NBD and one TMD) with one ATP-binding site. Some transporters carry only a single NBD or TMD are termed single-domain structures. In contrast, only NBDs are present at the N- and the C-terminus in non-transporter ABC proteins [7]. Based on the functional classification, ABC transporters are divided into three major classes; Classes 1 and 2 are involved in the translocation of substrates across the cell membrane, and Class 3 is mainly associated with DNA repair and regulation of gene expression in cells [8].

ABC transporters transport a wide range of chemical entities, including lipids, proteins, sugars, amino acids, xenobiotics, drugs and inorganic ions. Most of the family members act as transport proteins in which the ATP-binding domains bind and hydrolyse ATP to provide energy to translocate a variety of solutes across the biological membrane; however, the energising module is also involved in non-transport processes, for example, cellular processes [9,10]. In mammals, ABC transporters have been associated with the active efflux of drugs across membranes. This is often reflected by altered gene expression at the cellular level, for example, overexpression of ABC transporter genes leading to the increased drug efflux, which may occur in response to exogenous drug exposure. P-glycoproteins (P-gps) were the first active pump described for their overexpression in tumor cells responsible for multi-drug resistance [11]. Some ABC transporters that are associated with active efflux of anticancer drugs include ABCB1 (P-gps, MDR1), ABCC1 (Multidrug-resistance-associated protein; MRP1), ABCC2 (MRP2), ABCC3 (MRP3) and ABCG2 (breast cancer resistance protein; BRCP) subfamilies [12].

The involvement of ABC transporters in the development of anthelmintic resistance has also been identified in animal parasitic nematodes [13,14]; however, there is little information about the role of ABC transporters in plant parasitic nematodes. The ABC transporters have been associated with non-specific mechanisms of resistance, as they modulate the concentration of different drugs at drug target sites irrespective of the drug class. Although attempts have been made to investigate the domain and functionality of ABC transporter systems in parasitic organisms, only the mechanism of ABC transporters is well understood in nematodes. For example, ABC transporters reported in the free-living nematode *Caenorhabditis elegans* [4] act as efflux pumps and facilitate the ATP-dependent movement of xenobiotics, including drugs. Therefore, ABC transporters might be involved in the active efflux of anthelmintic drugs away from their target sites, resulting in decreased drug concentration and increased parasite survival (Figure 1). This has been shown by the increased efficacy of anthelmintic drugs in nematodes using multidrug resistance inhibitors (MDRIs), which inhibit the activity of ABC transporters [14,15]. In addition, it has been suggested that anthelmintics such as ivermectin (IVM), levamisole (LEV) and thiabendazole (TBZ) are substrates of ABC transporters [12,16], and there is considerable evidence that exposure to anthelmintics also modulates the expression patterns of different ABC transporters in nematodes [17,18]. In contrast, IVM treatment showed no effects on the expression patterns of ABC transporters in IVM-resistant *Haemonchus contortus* and *Cooperia oncophora* worms collected from treated animals compared to those collected from untreated animals [19,20]. This indicates that the expression patterns of ABC transporters in nematodes seem to be variable, with some studies linking them to anthelmintic resistance and other reports finding no association. Therefore, this review article provides a comprehensive overview of the current state of knowledge on the role of ABC transporters in the survival of parasitic nematodes, including the repertoire of ABC transporter genes, regulation of transporter expression, their physiological roles and the recent advancements in identifying and characterizing ABC transporters in parasitic nematodes. Additionally, the review will discuss the association of ABC transporters with anthelmintic resistance and the potential for targeting these transporters as a strategy for controlling parasitic nematodes.

## 2. ABC Transporters in Helminths

Helminths possess a larger number of ABC transporter genes than mammals, which have only a few multidrug resistance (MDR) transporters [21]. In contrast to mammals, only P-gps and multidrug resistance proteins (MRPs) are the members of ABC transporters that have been implicated in the development of resistance against anthelmintic drugs such as macrocyclic lactones (MLs) in parasitic nematodes [22,23,24], whereas the role of other members of the ABC transporter family is still unclear. Genes that are homologous to the mammalian ABCB1 subfamily encode P-gps in nematodes. Other MDR transporter genes include ABCC, which encodes MRPs, but half transporters are more closely related to the ABCB subfamily in nematodes than the ABCG2 subfamily in mammals [25].

Nematodes possess a greater diversity of ABC transporters compared to mammals; for example, *C. elegans* is known to have 15 P-gp genes (*pgp)*, eight MRP (*mrp*) and nine Half transporter genes (*haf*) [4]. Table 1 summarizes the diversity of ABC transporter genes that have been reported in parasitic nematodes. The reason for this diversity of ABC transporters in helminths is still not clear; however, it was opined that these might be essential for the protection of neurons in the worm body from a broad range of toxins [21].

All the stages of helminths do not express all ABC transporters, and their expression patterns may vary in different developmental stages. For example, Sarai et al. [43] described life-stage differences in expression patterns for various P-gp genes within isolates of *H. contortus and* the difference between the isolates. Therefore, the transport mechanism mediated through MDR transporters may vary in different developmental stages of nematodes. For example, Kerboeuf and Guegnard [16] reported that IVM failed to stimulate P-gps in eggs of resistant *H. contortus* isolate compared to other MLs. In contrast, Godoy et al. [44] showed that IVM was as effective as abamectin and more effective than moxidectin in inhibiting rhodamine-123 (R-123; a fluorescent substrate) transport through P-gps from adult worms of *H. contortus* expressed in mammalian cells. The difference in the interaction of anthelmintics across various life stages in nematodes also proposes that there might be variation in expression patterns of ABC transporters between developmental stages of nematodes. These studies suggest the important roles of ABC transporters for helminth survival; however, further work is required to better characterize these roles in the development of different helminth parasites.

## 3. Physiological Roles of ABC Transporters

As described earlier, ABC systems are divided into three major classes based on their functions; Classes 1 and 2 are involved in the translocation of substrates across the cell membrane and are found in prokaryotes and eukaryotes. Class 3 ABC systems do not function in the membrane and are mainly associated with cellular processes, for example, DNA repair, translation, and regulation of gene expression in cells [8]. Class 1 systems found in membranes act as importers, which were initially described in prokaryotes; however, recent studies have shown that these importers may also be found in protozoan parasites such as *Toxoplasma gondii* [45]. Class 2 systems are also present in membranes but function as exporters in prokaryotes and eukaryotes [46], reviewed by [47]. Although the ABC transport proteins in Class 3 do not act as transporters, they share a common structural component with ABC transporters, specifically the ATPase domain (also known as ABC-2), which utilizes the energy derived from ATP hydrolysis to recognize and bind mismatched DNA bases or DNA insertion loops [48]. Mammalian ABC transporters export several groups of molecules, for example, lipids, retinoic acid derivatives, cholesterol and sterols, bile acid, iron, nucleosides and peptides [49,50].

In nematodes, P-gps (ABCB1), one of the most studied ABC transporters, have been reported to play an important role in protecting nematodes against environmental toxins. The protective functions of several of these ABC transporters have been reported previously. For example, Pgp-3 protects *C. elegans* worms against natural toxins [51]. In addition, Issouf et al. [52] described the specific induction of *pgp-3* in *H. contortus* exposed to sheep eosinophil granules, suggesting a role in the detoxification of host immune cell products. Similarly, the expression of some P-gp genes (including *pgp-3* and *pgp-9*) in intestinal excretory cells of the closely related model organism *C. elegans* further suggests a role for them in protecting worms against toxic substances [13]. Similarly, in plant parasitic nematodes, upregulation of ABC transporter genes in *Bursaphelenchus xylophilus* was reported in response to α-pinene, a monoterpene produced by plants in response to attack [53]. In addition, the increased sensitivity of different P-gp knock-out strains of *C. elegans* to some anthelmintics, particularly IVM, provides evidence for their role in protection against anthelmintics [13,54]. Moreover, Luo et al. [55] suggested that tissue transcription and expression pattern of *Tc-abcg-5* may indicate an essential role for this transporter in the reproduction of *Toxocara canis*. Previously, we observed relatively higher expression levels of ABCF transporters (*abcf-1* and *abcf-2*) compared to many of the other genes in *H. contortus* [18]. ABCF transporters lack the transmembrane domains (TMDs) present in other transporter proteins, and their function as transporters is currently unclear. It has been suggested that ABCF transporters are involved in cell physiology (ribosome assembly, translational control and mRNA transport) in arthropods [56]. The absence of ABCF transporters in arthropods results in physical abnormalities, as shown by Broehan et al. [57], who observed the death of third-stage larvae (L_3_) and arrested growth, as well as the death as pharate adults in RNA interference studies with L_3_ and pupae of *Tribolium castaneum*, respectively. In *Schistosoma mansoni*, disruption of egg production following knockdown of *SMDR2* or *SmMRP1* expression using RNA interference further indicated an important role of ABC transporters in parasite reproduction [58]. However, the physiological role of these transporters in parasitic nematodes needs further investigation.

## 4. Methods of Studying ABC Transporters

Of the various ABC transporters found in nematodes, ABCB1 (previously known as P-gp) has been of particular interest due to its role in drug resistance and its potential as a therapeutic target. However, the identification/localization of ABCB1 in nematodes is challenging due to the complex nature of these organisms, with different life stages and protective structures. Confirming ABCB1 expression in different body tissues of nematodes requires a laborious and careful approach, and differentiation between ABCB1 and other ABC transporter proteins can be difficult. Sometimes, differentiation between P-gps and other proteins of the ABC transporters family is difficult due to the nature of methodologies employed or the abundance of other ABC transporters that may be found in every selected location [59]. Furthermore, ABCB1 isoforms that have not yet been characterized add to the challenge of identifying ABCB1 in nematodes [60]. Various methods, including molecular biology techniques, localization using monoclonal antibodies, biochemical assays and in vivo models, have been employed to detect ABCB1 proteins in nematodes. There have been various reports describing the presence of genes that encode different ABC transporters in nematodes, including *H. contortus* [26], *C. elegans* [4], *Brugia malayi* [29], *T. circumcinta* [32] and *O. volvulus* [38,39].

Several studies have reported the use of specific monoclonal antibodies for the detection of P-gps in nematodes. Different localisation studies have shown that ABC transporters are expressed in excretory cells, the intestine, amphids, neurons, muscles, pharynx, hypodermis, and other tissues, including the vulva in female worms [61]. The use of C219 and UIC2 monoclonal antibodies for detecting human and mouse *mdr-1* gene products has been documented, while in *H. contortus*, UIC2 also confirmed both the presence and activity of P-gps, as it was reported earlier in tumor cells [62,63]. Furthermore, P-gp expression has been shown to predominate in the gastrointestinal tract, particularly the posterior pharynx and anterior intestine in *H. contortus* [64], while anti-human *mdr1* monocloncal antibody staining showed that P-gps are also present in the eggshells [63] and cuticle of adult and larval stages of *H. contortus* [65]. In addition, David et al. [66] recently reported that *Hc-pgp-13* was expressed in pharyngeal, neuronal and epithelial tissues. Similarly, Chelladurai and Brewer [67] revealed the expression of *Peq*-P-gps in the intestine, body wall, nerves, lateral cords, and reproductive tissues of adult male and female *Parascaris equorum* worms. In *T. canis*, detection of tissue distribution and transcription of *TcABCG-5* showed that this ABC transporter was predominantly expressed in the reproductive tract of female worms, suggesting an essential role of ABCG-5 in the reproduction of this parasitic nematode [55].

Certain biochemical assays are available to detect P-gps that are based on measurement or inhibition of the activity of trans-membrane transport proteins and are classified as functional assays. Accumulation studies are based on the uptake of a radiolabeled or fluorescent probe into the cell in the presence of ABC transporter inhibitors, which block the efflux proteins leading to increased accumulation of the probe and evidence of the presence of these transport proteins. The studies in mammalian cells have been conducted using transfected cell lines that over-express a specific transporter protein of interest compared to wild-type cells [68]. Rhodamine-123 (R-123) is one of the most commonly used P-gp probes in mammalian tumors and transfected cells. Several agents identified as inhibitors of the transport proteins using efflux/accumulation assays have potential applications in drug resistance to chemotherapy in human cancers and livestock parasitic nematodes [16,69]. In nematodes, R-123 accumulation/efflux has been studied using eggs of *H. contortus* in the presence of P-gp inhibitors, especially verapamil. The results showed that P-gp inhibitors increase the accumulation of R-123 in the eggs of *H. contortus*, indicating partial or complete inhibition of drug efflux by these inhibitors [70,71]. ATPase assay is another biochemical tool capable of identifying the presence of specific transport protein channels by detecting the specific ATPase activity. The binding of ATP at a nucleotide-binding domain on the P-gps is crucial for substrate transport, followed by hydrolysis of ATP by P-gp-specific ATPase [72]. This assay requires a prepared cell membrane enriched with the efflux protein of interest, ATP, an analytical method to detect inorganic phosphate liberated from ATP hydrolysis and a mechanism for discriminating between general and P-gp specific ATPase activity. Some general ATPases, including Ca-ATPases, Na^+^ ATPases, K-ATPases and mitochondrial ATPases are inhibited by specific ATPase inhibitors [68]. Currently, ATPase assay has been most widely used for determining P-gp-mediated drug efflux transport in mammals, while no information is available about its use in nematodes. Studies in mammalian cells showed that various compounds, including verapamil, cyclosporine A, vinblastin and loperamide were able to stimulate P-gp ATPase activity [73,74].

Somechanged, in vivo methods are currently being used in mammalian studies to detect the presence of specific transport proteins. Transgenic animal models are a well-established means of evaluating genes and their protein products. Animals can be genetically modified such that a specific protein can be over-expressed or blocked by adding or deleting a gene(s). The phenomenon of the removal or silencing of a gene is called homologous recombination or, more commonly, gene knockout [68]. The role of P-gps in drug absorption and elimination has been studied by silencing single or both *MDR* genes (*mdr-1* and *mdr-2*) that encode P-gps in mice. The results showed increased absorption/accumulation of various drugs in different body tissues of mice lacking P-gps compared to the control group of mice [75,76]. The use of gene silencing to study the role of P-gps in nematodes is still limited and no success has been reported with parasitic nematodes. In the free-living nematode *C. elegans*, genetic modification has also been used as a tool to study the role of P-gps in anthelmintic sensitivity. Deletion of some P-gp genes in *C. elegans* resulted in increased sensitivity of the worms to IVM compared to wild-type worms [13]. However, this technique has not yet been successfully adapted for use with parasitic nematodes and needs further work.

## 5. ABC Transporters and Drug Resistance in Nematodes/Evidence for a Role in Resistance

### 5.1. Are They Substrates?

The absorption, distribution and elimination of the anthelmintics, especially MLs in hosts and parasites, are influenced by multidrug resistance transporters, including P-gps [59,77]. The combined effects of these efflux proteins have a considerable impact on the bioavailability and efficacy of the drugs by interfering with the absorption, distribution and elimination of anthelmintics [78]. The efficiency of drug elimination is higher with the increased affinity of anthelmintics for P-gps; therefore, anthelmintic drugs, including MLs, are eliminated from the organism more rapidly in relation to the relatively shorter resident time [77], which ultimately reduces the drug efficacy. There is accumulating evidence that anthelmintics act as a substrate for P-gps, and the interaction of MLs with P-gps is considerably better than the other anthelmintic groups [21]. Ivermectin was the first member of MLs reported as a substrate for P-gps when a recommended antiparasitic dose resulted in signs of toxicity and death of genetically engineered mice lacking the gene coding for P-gps [75]. The concentration of IVM was 100-fold higher in the brain of mutant mice than in wild-type mice. Apart from strong interaction with P-gps, MLs also interact with MRPs (MRP1, MRP2 and MRP3), suggesting that the efflux of MLs is influenced by multiple transporter proteins [79]. Likewise, the same mechanism of resistance might be inferred for other anthelmintic classes. Some evidence suggested that P-gps also utilize imidothiazole as their substrate, but LEV do not show any stimulatory effects; however, BZ derivative albendazole shows very slight stimulatory effects on the mammalian P-gps [80,81]. On the other hand, in nematodes, it has been shown that LEV and MLs other than IVM are the most P-gps-stimulating anthelmintics in eggs of a resistant isolate of *H. contortus* [16], as suggested by the poor transportation of IVM compared to other MLs. However, it was recently described that IVM markedly inhibited the rhodamine-123 transport through PGP-2, PGP-9.1 and PGP-16 from adult worms of *H. contortus* expressed in mammalian cells [44,82,83], whereas moxidectin showed lesser inhibitory effects on R-123 efflux. The authors concluded that IVM is a better substrate of nematode P-gps than moxidectin, and this may help to explain the slower rate of development of resistance to moxidectin compared with other avermectins in *H. contortus*. Similarly, our previous studies have shown that multidrug resistance inhibitors significantly increase the in vitro toxicity of IVM and LEV in *H. contortus*; and exposure to IVM and LEV induces transcription of various ABC transporter genes in *H. contortus* larvae [14,18]. In addition, pre-exposure of *H. contortus* larvae to IVM and LEV also increased the efflux of rhodamine-123, suggesting the stimulatory effects of these anthelmintics on ABC transporters.

### 5.2. Constitutive Expression of ABC Transporters and Anthelmintic Resistance

P-gps have been implicated in resistance to anthelmintics, with several studies describing an increased transcription of specific transporter genes in drug-resistant nematodes compared to their counterparts (Table 2). For example, studies on *H. contortus* have shown that multi-drug resistance is associated with increased P-gp mRNA expression levels in IVM-resistant strain [64]. Williamson et al. [84] reported significantly increased expression levels of *pgp-2* and *pgp-9* in a multi-drug resistant isolate compared to a susceptible isolate of *H. contortus*. Our previous studies also showed that *pgp-1*, *pgp-9.1* and *pgp-9.2* expressed at significantly higher levels in drug-resistant larvae of *H. contortus* compared to the susceptible larvae [18,43]. Similarly, in *T. circumcincta*, *pgp-9* showed significantly higher transcription in multidrug-resistant isolate compared to susceptible isolate [29]. In addition, expression levels of *haf-9* and *mrp-1* were also higher in the eggs of IVM-resistant *C. oncophora* [17]. In addition, Figueiredo et al. [85] recently reported that IVM-resistant *C. elegans* isolate showed upregulation of *pgp-12* and *pgp-13* with downregulation of all other P-gp genes, whereas Yan et al. [86] reported that mRNA levels of five ABC transporters were significantly higher in IVM resistant isolate of *C. elegans* compared to the wild-type susceptible isolate. On the other hand, Williamson and Wolstenholme [37] observed no significant changes in P-gp mRNA expression levels in a rapidly selected IVM-resistant isolate as compared to the drug-sensitive parent isolate. This suggests that the expression patterns of P-gps in nematodes seem to be variable, with some reports linking them to anthelmintic resistance and other studies finding no association.

### 5.3. Effect of Anthelmintics on ABC Transporters Expression

Since anthelmintics are substrates of P-gps, they may regulate the expression level of P-gps through transcriptional or post-transcriptional mechanisms [94]. Ardelli and Prichard [13] studied the effects of IVM on the expression patterns of P-gps and they found that IVM induces changes in both the amplitude and occurrence of 15 P-gp genes in *C. elegans,* whereas the inactivation of certain *pgps* (*pgp*-2, *pgp*-5, *pgp*-6, *pgp*-7, *pgp*-12 and *pgp*-13) resulted in increased sensitivity to IVM compared to the wild-type *C. elegans*. There is also accumulating evidence that exposure to anthelmintics increases gene transcription for ABC transporters in parasitic nematodes (Table 3). For example, previous studies reported overexpression of *pgp-11*, *pgp-16* and *mrp-1* in *C. oncophora* recovered from animals treated with IVM [17,95]. Similarly, in vitro exposure of *H. contortus* larvae of resistant and susceptible isolates to IVM, LEV, and monepantel increased the expression patterns of multiple ABC transporters [18,24]. This up-regulation of P-gps and other ABC transporters in nematodes due to anthelmintics exposure may lead to multiple-drug resistance in the nematode population. For example, the in vitro studies demonstrated that monepantel exposure resulted in increased tolerance of a proportion of the larval population to IVM and LEV. Similarly, Lloberas et al. [96] reported that treatment of infected lambs with IVM increased the transcription levels of *pgp-2* in resistant *H. contortus* compared to the worms collected from untreated control animals. In addition, the combination of P-gp interfering or MDR-reversal agents with IVM potentiated sensitivity to IVM in adults and microfilariae of *B. malayi* following overexpression of multiple ABC transporter genes in response to IVM exposure [97]. Recently, Diao et al. [98] reported that the expression of multiple *Bx-MRP* genes increased in the pine wood nematode, *Bursaphelenchus xylophilus*, with increasing concentrations of emamectin benzoate, avermectin and matrine, suggesting a potential role in the regulation of multidrug resistance.

Furthermore, De Graef et al. [17] also reported a significant increase (3–5 fold) in transcription levels of *pgp-11* in *C. oncophora* adult worms 14 days after treatment with IVM or moxidectin compared to unexposed adults. The authors further described a 4-fold transcriptional up-regulation of *pgp-11* in L_3_ of resistant isolate compared to susceptible L_3_ after 24 h in vitro exposure to different IVM concentrations (8.7 ng/mL and 87 ng/mL). In contrast, no significant differences were observed between the expression patterns of P-gp genes before and after IVM treatment in *Parascaris univalens*, *C. oncophora* and *H. contortus* [19,37,99]. Therefore, the role of these trans-membrane proteins as a drug efflux mechanism in nematodes is still unclear and needs further investigation.

**Table 3 pathogens-12-00755-t003:** A summary of the selected studies reporting the induced expression patterns of ABC transporters in parasitic helminths in response to anthelmintics exposure.

Parasite/ Helminth	Drug Selection	Significantly Differential Expression of ABC Transporters	Life Stage	References
		Resistant Isolate vs. Susceptible Isolate		
*B. malayi*	IVM	Higher expression of multiple P-gp and MRP genes in response to IVM exposure	Microfilariae and Adults	[97]
	MOX	Higher expression of multiple ABC transporter genes in response to moxidectin exposure	Adult Worms	[100]
*C. oncophora*	MLs	Increased transcription of *Con pgp-11* in resistant isolate only	L3	[87]
	IVM		L3 and adults	
Cyathostomins	IVM	Higher expression of *pgp-9* in resistant isolate as compared to the susceptible population	L3	[101]
*D. immitis*	IVM	Increase in gene expression for *Dim-pgp-10* and *DIM-pgp-11*	Adults	[102]
*E. granulosus*	Amiodarone Loperamide	*Eg-pgp1* and *Eg-pgp2* transcripts were up-regulated in response to in vitro drug treatment	Protoscoleses & metascoleces	[31]
*F. gigantica*	Taurocholate	Increased expression of *MDR1* and *MRP1*	Adult flukes	[33]
	Triclabendazole	Increased expression of *MRP1*		
*H. contortus*	IVM	*In-vivo* IVM exposure significantly increased *pgp-3* and *pgp-9.2* transcription in resistant isolate	Adults	[90]
	IVM	*In-vivo* IVM exposure significantly decreased *pgp-3* and *pgp-9.1* transcription, whereas *pgp-2* expression was progressively increased in IVM-resistant isolate	Adult worms	[103]
	IVM	Resistant isolate: Increased expression of *pgp-1, pgp-2, pgp-9.1, pgp-10, pgp-11* and *haf-6.*	L3	[18]
	LEV	In resistant isolate: Upregulation of *pgp-1, pgp-2, pgp-9.1, pgp-10, pgp-11, abcf-1* and *haf-6 genes.* No significant changes in susceptible isolate	L3	
	MPL 250 µg/mL	In both MDR and susceptible isolates, expression of multiple ABC transporter genes increased at 3, 6 and 24 hrs. Some *pgps* were also decreased.	L3	[24]
	MPL 2.5 µg/mL	MDR isolate: Expression of multiple ABC transporter genes decreased. Susceptible isolate: *pgp-11* increased at 3 hr, while *pgp-9.1* and *pgp-11* decreased at 24 hr.		
	IVM	No changes in P-gp expression levels in IVM-resistant isolate compared to drug-susceptible parent	L3	[37]
*S. mansoni*	PRAZ	Transient increase in transcription levels of *SMDR2* in schistosome isolate with reduced praziquantel susceptibility	Adult worms	[93]
	PRAZ	Transient increase in transcription levels of *SmMRP1* following exposure of worms to sub-lethal concentrations of praziquantel	Juvenile adults	[104]

L3 = third-stage larvae, L4 = fourth-stage larvae, MDR1 = multidrug resistance protein 1, MRP = multidrug resistance-associated protein, MLs = Macrocyclic lactones, IVM == Ivermectin, MOX = Moxidectin, LEV = Levamisole, MPL = Monepantel, PRAZ = Praziquantel.

### 5.4. Polymorphism in ABC Transporters and Anthelmintic Resistance

In addition to transcriptional upregulation, allelic polymorphisms in ABC transporters can potentially increase drug efflux from the cells, thus changing the drug distribution within the parasite’s tissue and moving anthelmintics away from target sites [21]. Table 4 summarizes the studies reporting constitutive and induced genetic polymorphisms in ABC transporters in parasitic helminths. For example, MLs select for certain alleles of P-gps as evidenced by changes in genetic polymorphism in P-gp A of *H. contortus* following selection with IVM and moxidectin [22]. Similarly, benzimidazoles have also been reported for selecting specific alleles of P-gps (allele P) in *H. contortus* [105]. Furthermore, polymorphism in *Tci-pgp-9* has also been reported in multidrug-resistant *T. circumcincta* [29]. Choi et al. [106] also described alternative splicing and four non-synonymous, exonic SNPs in *Tci-pgp-9* in a multidrug-resistant *T. circumcincta* strain compared to the “near-isogenic” drug-susceptible sister strain. Recently, Turnbull et al. [107] supported the previous findings and reported that the non-synonymous nucleotide substitutions in *Tci-pgp-9* sequence variants shared by the multiple-resistant UK and New Zealand isolates were not observed in their drug-susceptible counterpart. These studies suggest that increased expression of *Tci-pgp-9,* as well as higher sequence polymorphism may play an essential role in anthelmintic resistance in *T. circumcincta.*

Furthermore, specific SNPs have also been reported in *D. immitis* resistant populations with reduced efficacy to MLs [30]. The authors suggested that a few of these SNPs may affect protein expression or function, substrate specificity and resistance development. Similarly, in human parasitic nematodes, polymorphisms associated with IVM selection have been reported in ATP binding sites of two ABC transporter genes in *O. volvulus*. However, the samples collected in 2002 showed less genetic variability than those collected in 1999, and the reasons for this still need to be clarified [108]. The authors suggested that the widespread use of IVM in Ghana for the control of onchocerciasis may be exerting selection pressure and reducing genetic variability. Although further studies are required to confirm the association of allelic polymorphism in ABC transporters with anthelmintic resistance, the previous studies indicate that anthelmintic selection pressure leads to increased genetic variability in ABC transport proteins.

**Table 4 pathogens-12-00755-t004:** Constitutive and induced genetic polymorphisms in ABC transporters in parasitic helminths.

Parasite/Helminth	Anthelmintic Drug	Genetic Polymorphism in ABC Transporters	Life Stage	References
		Resistant Isolate vs. Susceptible Isolate		
*C. oncophora*		Multiple SNPs led to different amino acid sequence variations in resistant isolates compared to susceptible isolates.	L3	[87]
*D. immitis*	MLs	The alternate allele frequency of the *D. immitis pgp-11* SNP marker ranged from 36% to 40% in resistant isolate and from 0% to 12% in susceptible isolate	MF	[88]
		75 SNPs and 89 SNPs in 15 ABC transporter genes of resistant and susceptible isolates, respectively	MF	[30]
*H. contortus*	BZs	Multiple allelic polymorphisms in *pgp-A* were detected against the cambendazole-selected strain of *H. contortus*, derived from the sensitive strain	L3	[105]
	IVM and MOX	Multiple allele variation in P-gp locus between the drug selected and susceptible strains	L3	[22]
*O. volvulus*	IVM	Resistant isolate: Reduced polymorphism in *OvPLP*, *OvMDR-1*, *OvABC-1*, *OvABC-3* and *OvPGP* Susceptible isolate: Polymorphism not detected	Adult	[109]
*P. equorum*	MLs	Three SNPs causing missense mutations in the *PeqPgp-11* were correlated with reduced sensitivity to MLs	Eggs	[92]
*T. circumcincta*	IVM	Nine non-synonymous SNPs in *Tci-pgp-9* in the MDR isolate sequences relative to the susceptible isolate	L3	[107]
	IVM	Alternative splicing and four non-synonymous, exonic SNPs in *Tci-pgp-9* gene in MDR isolate compared to susceptible “near-isogenic” sister strain.	L3	[106]

L3 = third-stage larvae, L4 = fourth-stage larvae, MDR = multidrug resistant, MF = Microfilariae.

## 6. ABC Transporters as Potential Targets to Control Nematodes

ABC transporters are critical in maintaining cellular homeostasis and essential for many physiological processes. Due to their broad substrate specificity and importance in cellular function, ABC transporters have become an attractive target for developing new antiparasitic drugs. For example, the characterization of ABC transporter genes may contribute to identifying gene targets for silencing and provide novel strategies for nematode control. In addition, the efficacy of existing drugs can be enhanced to overcome drug resistance in nematodes by modulation of ABC transporters.

### 6.1. Genetic Manipulation/Gene Silencing

Gene silencing at the gene expression level of ABC transporters using anti-sense oligonucleotides or double-stranded small interference RNAs (siRNA) to regulate mRNA levels or target the signaling pathways that induce ABC transporters expression has been reported to counter multi-drug resistance in mammalian cancer research [110,111]. Similarly, in nematodes, there have been reports describing the use of gene silencing; for example, *mrp-1* and *pgp-1* knock-down in *C. elegans* led to increased sensitivity to drugs or heavy metal ions compared to the wild-type [112]. Furthermore, silencing of some P-gp genes in *C. elegans* increases the sensitivity of genetically modified worms to MLs compared to wild-type worms [13,54]. The use of RNAi to knock down some specific genes has had limited success with *H. contortus*, as described by some of the previous studies [113,114,115]. In addition, no information is available as to whether it can silence transporter genes in parasitic nematodes. Therefore, further research is required to elucidate the use of these high-throughput techniques to counter anthelmintic resistance in parasitic nematodes.

### 6.2. Modulation of ABC Transporters

The studies reporting non-specific mechanisms of anthelmintic resistance in nematodes show that P-gps are the only member of the ABC transporter family that is well understood for its role in anthelmintic resistance. The research directions to overcome the MDR in nematodes are still less advanced, and further work is required to counter this mechanism. Increased efficacy of anthelmintics against parasites could result from modifying the pharmacokinetics in the host or by blocking the resistance-conferring transport mechanism in parasites [25]. Therefore, the approach more commonly adopted in mammalian and nematode research is identifying and developing an effective MDR inhibitor (MDRI) or reversal agent [12]. The idea of the modulation of P-gps in nematodes to reverse drug resistance partially or completely is based on the principal mechanism of action of these transporters. These transport proteins reduce drug toxicity by transporting the drug away from its target site. Therefore, reducing the activity of transporters by using such compounds, which either block these channels or compete with anthelmintics, would increase drug toxicity (Figure 2).

Ideally, an effective MDRI would be a non-toxic, potent and specific inhibitor of the relevant ABC transporters and would have no adverse effects on the pharmacokinetics of anthelmintic agents [12]. Several compounds have been evaluated in vitro and in vivo in mammals, while some of them have also been studied in nematodes. These compounds are classified into three different generations of ABCB1 inhibitors. Previous studies which have reported the effects of MDR inhibitors on the sensitivity of parasites to anthelmintics are summarized in Table 5.

#### 6.2.1. First-Generation MDR Inhibitors

The first-generation inhibitors were developed for some other use, such as verapamil (anti-hypertensive), cyclosporine A (immunosuppressant) and quinine (antimalarial). Verapamil is a well-studied multidrug resistance inhibitor and has been shown to inhibit mainly the functions of P-gps in mammalian tumor cells and nematodes [12,59]. However, in mammals, these inhibitors failed to provide positive effects in clinical studies, despite their in vitro efficacy in inhibiting ABCB1 transporters. These agents were reportedly toxic and may induce pharmacokinetic complications (cardiac toxicity in the case of verapamil) in mammals [125].

In nematodes, it has been shown that verapamil can reverse the anthelmintic resistance either partially or completely when co-administered with anthelmintics [70,71,97]. Verapamil has been studied in various in vitro assays such as egg hatch assay (EHA), larval development assay (LDA) and larval migration inhibition assay (LMIA) using different developmental stages (eggs and L3) of trichostongyloid nematodes [13,14,70,97]. The results showed that co-administration of verapamil increases the sensitivity of both the susceptible and resistant isolates of *H. contortus* and *C. oncophora* to anthelmintic agents by decreasing the IC_50_ values compared to anthelmintics alone. Verapamil also increased the thiabendazole (TBZ) toxicity in EHA and showed a partial reversal of the resistance [70]. Interestingly, Raza et al. [14] showed that in LDA, verapamil increased IVM toxicity to a resistant isolate of *H. contortus* only, whereas it increased LEV toxicity to both resistant and susceptible isolates and showed no effect on IC_50_ values of thiabendazole. In contrast, using verapamil in larval migration assay (LMA) did not show any effects on the toxicity of IVM and LEV. The authors suggested that these patterns could be attributed to variations in life-stage dependent expression levels of specific P-gp genes between early larval life stages (examined in LDA) compared to later infective-stage larvae (as examined in LMA). Furthermore, some in vivo studies showed that verapamil, when co-administered with anthelmintic agents, increased the bioavailability of IVM in jirds and sheep [126,127]. In addition, verapamil and cyclosporin A increased the IVM sensitivity of *H. placei*-resistant isolate in LMIA [122]. The investigations of verapamil toxicity are still in the early stages in livestock; for example, Pérez et al. [128] reported that the co-administration of verapamil/IVM in pregnant sheep increases the bioavailability of IVM not only in maternal blood but also in fetal blood which may lead to IVM toxicity. Therefore, further research is required.

#### 6.2.2. Second-Generation MDR Inhibitors

This group of MDR inhibitors were designed to counter the major drawbacks of first-generation inhibitors (reduced specificity and increased toxicity). Valspodar, a derivative of cyclosporine A, is characterized by higher in vitro specificity and potency than its precursor and no immunosuppressive effects, but it failed to improve the outcome of phase II clinical trials when administered with anticancer agents. Valspodar also inhibited cytochrome P450 (CYP450), which led to higher systemic concentrations of both the inhibitor and the therapeutic drug resulting in increased toxicity [129,130]. Biricodar, derived from piperidine was also a more potent ABCB1 inhibitor than the first-generation compounds and showed the ability to inhibit the ABCC1 transporter family. However, biricodar showed no efficacy in phase II clinical trials when co-administered with doxorubicin or vincristine in addition to the significant complication of neutropenia [131].

Information on the role of second-generation inhibitors in nematodes is limited. Bartley et al. [121] reported that using valspodar in combination with IVM significantly increases the in vitro sensitivity of drug-susceptible and -resistant isolates of *H. contortus* and *T. circumcincta* in larval feeding inhibition assay. The authors also suggested that the combination of P-gp inhibitors with drugs could be useful to counter the emergence of anthelmintic resistance either by increasing the drug’s efficacy or by shortening the course of treatment in livestock. Valspodar has also been reported to reverse the resistance in IVM-selected *C. elegans* (free-living nematode) isolate. This isolate also showed increased expression of P-gps and MRPs following IVM selection, and reversal of resistance on the addition of valspodar clearly suggested that it interferes with the functions of ABC transporters, thus ultimately reversing the resistance [132]. In addition, valspodar also showed increased IVM, LEV and TBZ toxicities to *H. contortus* larvae in LDA and increased IVM toxicity in LMA. This increased efficacy of valspodar was more marked in LMA for IVM with both resistant and susceptible isolates (up to 4.5-fold increased IC_50_ values) [14]. However, studies with adult parasites would be required to assess the practical applicability of the ML-resistance reversing ability of this compound since this is the target life stage of most chemotherapeutic approaches to worm control.

#### 6.2.3. Third-Generation MDR Inhibitors

The third-generation MDR inhibitors were specifically designed to counter the limitations of first two generation inhibitors; therefore, inhibitor development was focused on the compounds that avoided the inhibition of CYP450 and did not alter the pharmacokinetics of the drugs. The members of third-generation inhibitors include tariquidar (an anthranilamide, XR9576), elacridar (an acridone caroxamide), zosuquidar (LY 335979), CBT-1 (quinolone derivative) and laniquidar (piperidine) [12]. They have higher potency and selectivity and lower toxicity than the agents of the previous two generations. However, the combination of zosuquidar and tariquidar with anticancer drugs (athricycline, taxanes, or docetaxel) showed little additional benefits [133,134], but these compounds did not show any toxicity.

There are limited reports available on the use of third-generation inhibitor tariquidar in pathogenic parasites. Kasinathan et al. [58] showed that tariquidar reduces egg production in *Schistosoma mansoni* in vitro and in vivo. It eliminated egg production in vitro at a concentration of 12.5 µM. The reduction in egg production was due to the association of MDR transporters with normal cellular physiology, evidenced by the disruption of parasite egg deposition in worms due to *SMDR2* and *SmMRP1* genetic knockdown. In *H. contortus*, the use of third-generation inhibitors, zosuquidar and tariquidar rendered the resistant larvae more sensitive to IVM than susceptible larvae [14]. Interestingly, though, while tariquidar significantly affected the IVM IC_50_ for both susceptible and resistant isolates, zosuquidar only reduced the IC_50_ for resistant larvae. These inhibitors resulted in a 5–6-fold increase in IVM toxicity, highlighting the potential usefulness of combination therapy (anthelmintics and MDRIs) in restoring the sensitivity of resistant worms (resistant-breaking strategy) and reducing the recommended dose of an anthelmintic while maintaining 100% efficacy against susceptible worms.

#### 6.2.4. Nutraceuticals as MDR Inhibitors

Rising levels of anthelmintic resistance stimulated significant research into the principle of integrated parasite management to complement regular drug treatment. One alternative control option is the use of nutraceuticals or bioactive forages containing bioactive compounds which can antagonize enteric nematodes [135]. Such active compounds may include sesquiterpene lactones, found in chicory, as well as polyphenolic compounds such as flavonoids or condensed tannins (proanthocyanidins) found in numerous plants such as sainfoin or birdsfoot trefoil [136,137,138]. The mode-of-action of these natural compounds is thought to mainly derive from the pharmacological-like binding of these bioactive plant metabolites to the worms, resulting in mortality and expulsion, but more indirect effects through boosting of the host immune system or modulation of the host gut microbiome have also been proposed [139,140]. It is becoming increasingly appreciated that drug pharmacokinetics can be markedly influenced through interactions with diet and gut microbiota factors [141]. Thus, it has been suggested that some of these plant metabolites may be able to modulate P-gp or other ABC transporters and thus influence anthelmintic drug pharmacokinetics and/or efficacy. For example, many bioactive plants, such as sainfoin are rich in polyphenols with a similar structure to quercetin-a well-known P-gp inhibitor [142]. Many polyphenols have been investigated for their health benefits in humans, e.g., by modulating transporter activity and thereby overcoming MDR in cancer [143,144,145]. Dupuy et al. [146] proposed the use of dietary quercetin to lower the rate of moxidectin metabolism and excretion by the host, and thus result in prolonged bioavailability and hence exposure of parasites to the drug. Furthermore, direct interactions between polyphenols and worm P-gp could result in drug accumulation within the parasite and reversal of drug resistance, which arises from increased P-gp expression. Direct evidence of this is still lacking, although in vitro evidence has shown that *H. contortus* and *Ascaris suum* may be more susceptible to IVM and LEV following exposure to polyphenols [147,148]. Whether this results from altered transporter activity (and hence higher drug concentrations in the worm) or other mechanisms, such as reduced cuticle integrity, is not clear and warrants investigation. Interestingly, in vivo evidence also exists that IVM treatment against nematode infection in lambs is more effective when the animals consume polyphenol-rich redberry juniper [149]. However, Gaudin et al. [150] reported that IVM treatment was less effective in lambs fed sainfoin, emphasizing the complexities of these drug-nutraceutical-parasite interactions in the in vivo situation. However, given that bioactive dietary additives are a rapidly growing tool to promote animal health and reduce enteric infections, opportunities to leverage the bioactivities of natural dietary compounds to modulate nematode drug transporters should be further explored.

## 7. Conclusions and Recommendations

ABC transporters play a crucial role in regulating the transport of various chemical entities in nematodes. The ABC transporter superfamily is diverse in structure and function, with some members acting as transporters for a wide range of substrates, including anthelmintic drugs, while others are associated with cellular processes such as DNA repair and gene regulation. In nematodes, ABC transporters have been implicated in developing anthelmintic resistance by functioning as efflux pumps, reducing drug concentration at their target sites. The activity of these transporters can be inhibited by MDRIs, leading to increased efficacy of anthelmintic drugs. Repurposing of existing drugs can save significant time and resources in drug development and may lead to the discovery of novel mechanisms that can be exploited in combination with existing anthelmintics. Although numerous MDR inhibitors have been developed, most of these have limited efficacy, high toxicity or have shown poor efficacy in vivo. Therefore, developing novel MDR inhibitors with improved potency and safety profiles is essential for improving clinical outcomes against parasitic nematodes. For example, natural products/nutraceuticals have been considered as important sources of new drugs due to their high biodiversity, oral bioavailability, and relatively low intrinsic toxicity. Therefore, natural products/nutraceuticals are appealing candidates for combining with chemotherapy to increase the cytotoxic effects of chemotherapeutic agents and reverse MDR. However, further research is required to fully understand the role of ABC transporters in parasitic nematodes and their potential as targets for controlling parasitic nematodes.

## Figures and Tables

**Figure 1 pathogens-12-00755-f001:**
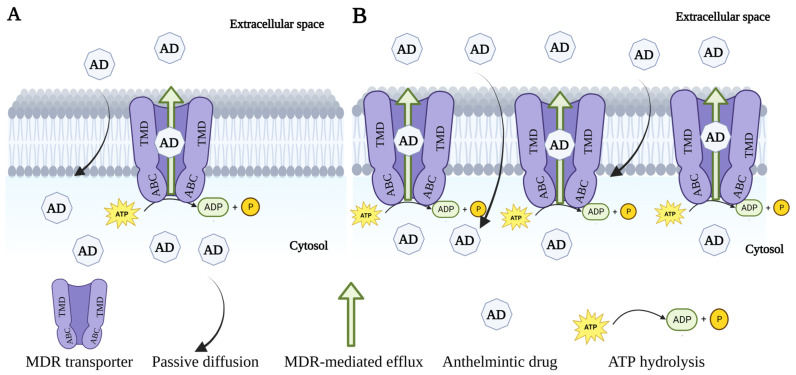
Multidrug resistance transporter-mediated efflux of anthelmintic drugs and their association with anthelmintic resistance. (**A**) The constitutive expression of MDR transporters in cell membranes indicates the basal efflux of anthelmintic drugs. (**B**) Overexpression of MDR transporters in response to substrates, such as anthelmintic drugs, leads to increased efflux of drugs and the development of resistance.

**Figure 2 pathogens-12-00755-f002:**
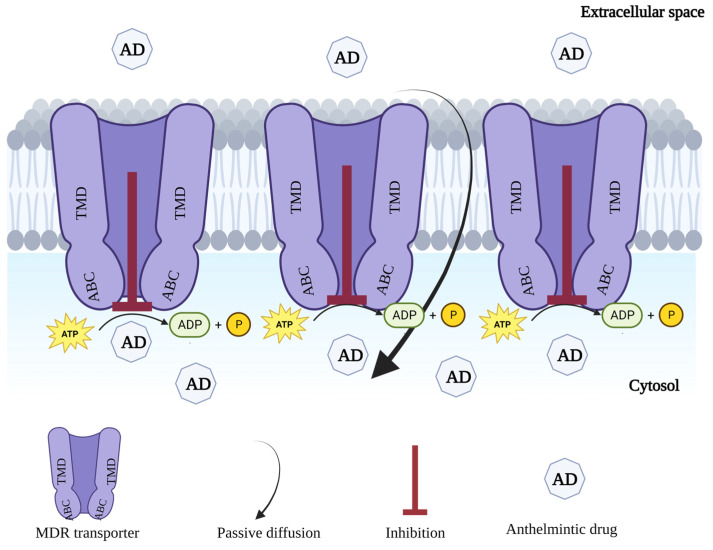
Schematic diagram of inhibition of ABC transporter/MDR transporter-mediated efflux of anthelmintic drugs in helminths with the addition of MDR reversal agent, which results in increased drug concentration and toxicity into the target cells.

**Table 1 pathogens-12-00755-t001:** Summary of reported ABC transporter genes in parasitic nematodes.

Parasites	ABC Transporter Genes	References
*Brugia malayi*	8 *pgps*, 5 *mrps*, 8 *haf* genes	[26]
*Bursaphelenchus xylophilus*	106 ABC transporter genes	[27]
*Cooperia oncophora*	7 *pgps*, 3 *mrps* and 5 *haf* genes	[17]
Cyathostomins	2 *pgps* genes	[28]
*Dirofilaria immitis*	3 *pgps*, 2 ABC-B (*Dim-haf-1*, and *Dim-haf-4*) and 2 ABC-C (*Dim-haf-5.1* and *Dim-haf-5.2*), 1 pseudogene	[29]
	3 *pgps*, 2 ABC-A (*Dim-abt-2*, *Dim-abt-4*), 5 ABC-B (*Dim-haf-1 Dim-haf-4*) and 5 ABC-C (*Dim-mrp-1*, *Dim-mrp-5*, *Dim-mrp-7*, *Dim-haf-5.1 and Dim-haf-5.2*), 2 ABC-G transporters (*Dim-wht-4*, *Dim-wht-4*), 1 pseudogene	[30]
*Echinococcus granulosus*	5 *pgp* genes	[31]
*Teladorsagia circumcincta*	11 *pgps* genes (partial sequences)	[32]
*Fasciola gigantica*	4 ABC transporters (*MDR1*, *MRP1*, *BCRP*, and *BSEP*) genes	[33]
*Fasciola hepatica*	1 P-gp orthologue gene	[34,35]
*Haemonchus contortus*	11 *pgps*, one *haf*, two *mrps* and 2 *abcf* genes	[36,37]
*Onchocerca volvulus*	2 *pgps and* 1 *haf* genes	[38,39]
*Opisthorcis felis*	4 ABC-A, 8 ABC-B (including 4 *pgps*), 6 ABC-C, ABC-D, 2 ABC-F, 3 ABC-G transporter genes	[40]
*Schistosoma mansoni*	20 ABC transporter genes, including *pgps* and *mrps*	[41]
*Toxocara canis*	1 ABC-B and 1 ABC-C transporter genes	[42]

**Table 2 pathogens-12-00755-t002:** A summary of the selected studies reporting the constitutive expression patterns of ABC transporters in parasitic helminths.

Parasite/Helminth	Differential Expression of ABC Transporters	Life Stage	References
*C. oncophora*	Constitutive overexpression of *Con haf-9* and *mrp-1* in resistant isolate as compared to susceptible	Eggs	[87]
*D. immitis*	A significant lower level of *DimPgp-11* constitutive expression in the ML-resistant JYD-34 isolate compared to the ML-susceptible Missouri isolate	MF	[88]
*H. contortus*	Significantly higher transcription level of *P-gp-9.1* in Ivermectin resistant strain as compared to the susceptible isolate	Adults	[89]
	Higher expression of *pgp-16* in resistant isolate	Eggs	[90]
	Higher expression of *pgp-2 and pgp-10* in resistant isolate	L4	[91]
	Higher expression of *pgp-1, pgp-9, pgp-12, pgp-14*, and *pgp-16* in resistant isolate	Adults	
	Constitutive overexpression of *pgp-1*, *pgp-9.1* and *pgp-9.2* in multi-drug resistant isolate compared to susceptible isolate.	L3	[18]
	Significant overexpression of *Hco-pgp-3*, *Hco-pgp-9.2*, *Hco-pgp-11* and *Hco-pgp-16* transcripts in L4 and adults as compared to free living stages	L3, L4 and Adults	[52]
	Significantly increased transcription of *Hco-pgp-2*, and *Hco-pgp-9* in resistant and *Hco-pgp-1* in susceptible cattle	L3	[84]
*P. equorum*	No difference	Eggs	[92]
	Significant overexpression of *pgp-11* in resistant worms	L3	
*S. mansoni*	Transient increase in transcription levels of *SMDR2* in schistosome isolate with reduced praziquantel susceptibility	Adult worms	[93]
*T. circumcincta*	Constitutive overexpression of *Tci-pgp-3, Tci-pgp-a and Tci-pgp-9* in multidrug-resistant isolate compared to susceptible isolate.	Eggs	[32]
	Constitutive overexpression of *Tci-pgp-9* and *Tci-pgp-e* in multidrug-resistant isolate compared to susceptible isolate.	xL3	
	Constitutive overexpression of *Tci-pgp-9* and *Tci-pgp-d* in multidrug-resistant isolate compared to susceptible isolate.	Adults	

L3 = third-stage larvae, xL3 = ex-sheathed third -stage larvae, L4 = fourth stage larvae, ML = Macrocyclic lactone, MF = Microfilariae

**Table 5 pathogens-12-00755-t005:** Effects of multidrug resistance inhibitors on the sensitivity of parasites to anthelmintics.

Nematode spp.	Inhibitors Used	Study Design	Effect on Anthelmintic Efficacy	Reference(s)
*A. pegreffii*	Valspodar, MK-571	In vitro	Increased toxicity of LEV, Nerolidol and Farnesol	[116]
*B. malayi*	Verapamil, cyclosporin A, vinblastine, and daunorubicin	In vitro motility assay	Increased susceptibility of adult and microfilariae to IVM	[90]
*C. oncophora*	Verapamil	In vitro LDA, LMIA	Completely restored sensitivity of IVM-resistant isolate	[97]
Cattle nematodes	Verapamil	In vitro EHA, LDA, LMIA	Increased IVM sensitivity	[117]
Cattle nematodes	Loperamide	In vivo	Increased IVM and MOX efficacy in terms of reduced FEC	[118]
*H. contortus*	Verapamil, Limonene and other phytochemicals	In vitro LDA, AMT	Verapamil and Limonene completely restored IVM sensitivity in resistant isolate	[119]
	Verapamil	In vitro LDA	Increased the susceptibility of wild-type and ML-selected isolates to IVM and MOX	[120]
	Crizotinib	In vitro LDA and LMA	Increased IVM toxicity to resistant isolate in LMA, and both resistant and susceptible isolates in LDA	[15]
	Verapamil. valsopodar, elacridar, zosuquidar, tariquidar	In vitro LDA and LMA	Increased the sensitivity of both resistant and susceptible isolates with comparatively marked effects in resistant isolate.	[14]
	Verapamil	In vitro (EHA)	Increased sensitivity of resistant isolate to thiabendazole	[71]
	Verapamil	In vitro (EHA)	Increased BZ sensitivity of resistant and susceptible isolates	[70]
*H. contortus*, *T. circumcincta*	Valspodar, verapamil, quercetin, ketoconazole, pluronic acid P85	In vitro (LFIA)	Significantly increased IVM sensitivity of susceptible and resistant isolates	[121]
*H. placei*	Cyclosporin A, ceftriaxone, dexamethasone, diminazene aceturate, quercetin, trifluoperazine, verapamil, vinblastin	In vitro (LMIT)	All inhibitors increased IVM sensitivity of resistant isolate, except diminazene aceturate	[122]
*S. mansoni*	Elacridar, tariquidar, zosuquidar, verapamil	In vitro (worm motility)	Significantly increased sensitivity of adult worms to praziquantel	[123]
*S. mansoni*	Cyclosporin A, dexverapamil, curcumin derivative (C-4), tariquidar, MK-571	In vitro and In vivo in mice	In vitro and in vivo disruption of egg production in resistant isolate	[58]
Sheep nematodes	Loperamide	In vivo in sheep	Increased IVM efficacy in terms of reduced FEC, increased plasma and availability.	[124]

EHA = egg hatch assay; LFIA = larval feeding inhibition assay; IVM = ivermectin; MOX = moxidectin; ML = Macrocyclic lactones; BZ = Benzimidazole; LMIT = larval migration inhibition test; FEC = faecal egg count; LMIA = larval migration inhibition assay; LDA = larval development assay; *A = Anisakis; Brugia*; *C.* = *Cooperia*; *H.* = *Haemonchus*; *S.* = *Schistosoma; T* = *Teladorsagia.*

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
