# Peer review of "Importance of ABC Transporters in the Survival of Parasitic Nematodes and the Prospect for the Development of Novel Control Strategies"

_pathogens, 2023, doi:10.3390/pathogens12060755_

Round 1
Reviewer 1 Report
The authors wrote and submit a review on the importance of ABC transporters in survival of nematodes and the prospect for development of novel control strategies. This review is of interest for people working of nematodes and development of novel compounds.
L112 to the end of page 3: the paragraph should be easier to understand with data gathered within a table
L150 : the authors have to include class 3 and informations of this class within the two first sentences of the paragraph.
L161 : the authors have to explain what is ABC-2.
L181-182 : the authors explained that they have observed data but they miss to include the corresponding reference.
Why the beginning of paragraph 4 is only dedicated to P-gp ? the authors should add sentences to introduce this point. Moreover, P-gp have been nowadays since more than 10 years renamed ABCB1. This abbreviation should be used instead of P-gp.
Paragraph 5 : what is the abbreviation MLs which appear here for the first time ? same for LEVs on page 7.
Paragraph 5 : proteins must be written with uppercase and gene in lower case.
Table 1/2/3/4 : they contain a lot of interesting and relevant informations but are difficult to read. They lack of synthesis and could be simplified to help the readers. In table 3, drug should be written with abbreviation since they are used in the text. In table 4, the table must be improve by group informations depending on inhibitor or on nematode; moreover, the authors must suppress the additional line corresponding to the head of the table.
Minor revisions:
L41-42-43 : TMD instead of TBDs
L77 : change the verb typified which is not adequate
Figure 1: the quality of the figure should be improved and in particular the AD which is difficult to read.
L100-101 : this sentence should be reformulated since it’s difficult to understand if helminths has or not the same number of abc transporter genes as mammals.
All in vivo and in vitro terms should be written in italics
Page 7 : the efflux instead of ethe fflux
First sentence of paragraph 5.3 : since…, they may regulate instead of these may
Author Response
The authors thank the reviewers for their positive feedback and relevant comments.
Please note that the line numbers referred to in response to the reviewer's comments correspond to the line numbers in the tracked changes version of the manuscript.
Response to Reviewer 1:
The authors wrote and submit a review on the importance of ABC transporters in survival of nematodes and the prospect for development of novel control strategies. This review is of interest for people working of nematodes and development of novel compounds.
- L112 to the end of page 3: the paragraph should be easier to understand with data gathered within a table
Response: The information in this paragraph has been converted into Table 1, as suggested by the reviewer. The references have been rearranged in the bibliography list as well (Table 1)
- L150: the authors have to include class 3 and informations of this class within the two first sentences of the paragraph.
Response: The text has been modified as advised. (Line 164-166)
- L161: the authors have to explain what is ABC-2.
Response: The paragraph is modified as advised and is now read as “the ABC proteins in class 3 do not act as transporters; however, they share a common structural component with ABC transporters, specifically the ATPase domain (also known as ABC-2), which utilises the energy derived from ATP hydrolysis to recognise and bind mismatched DNA bases or DNA insertion loops [48].” (Line 172-176).
- L181-182: the authors explained that they have observed data but they miss to include the corresponding reference.
Response: Reference is now added. (Line 199)
- Why the beginning of paragraph 4 is only dedicated to P-gp? the authors should add sentences to introduce this point. Moreover, P-gp have been nowadays since more than 10 years renamed ABCB1. This abbreviation should be used instead of P-gp.
Response: The paragraph is modified as advised and is now read as “Of the various ABC transporters found in nematodes, ABCB1 (previously known as P-gp) has been of particular interest due to its role in drug resistance and its potential as a therapeutic target. However, the identification/ localisation of ABCB1 in nematodes is challenging due to the complex nature of these organisms, with different life stages and highly differentiated organs and protective structures. Confirming ABCB1 expression in different body tissues of nematodes requires a laborious and careful approach, and differentiation between ABCB1 and other ABC transporter proteins can be difficult. Sometimes differentiation between P-gps and other proteins of the ABC transporters family is difficult due to the nature of methodologies employed or the abundance of other ABC transporters that may be found in every selected location [59]. Furthermore, ABCB1 isoforms that have not yet been characterised add to the challenge of identifying ABCB1 in nematodes [60]. Various methods, including molecular biology techniques, localisation using monoclonal antibodies, biochemical assays, and in vivo models, have been employed to detect ABCB1 in nematodes.” (Lines 211-224)
- Paragraph 5: What abbreviation MLs appear here for the first time? same for LEVs on page 7.
Response: The term Macrocyclic lactones (MLs) was mentioned on page 3 (line 104), and levamisole was mentioned on page 2 (line 77).
- Paragraph 5: proteins must be written with uppercase and gene in lower case.
Response: Proteins are not written in Uppercase, and genes are in lowercase.
- Table 1/2/3/4: they contain a lot of interesting and relevant informations but are difficult to read. They lack of synthesis and could be simplified to help the readers. In table 3, drug should be written with abbreviation since they are used in the text. In table 4, the table must be improve by group informations depending on inhibitor or on nematode; moreover, the authors must suppress the additional line corresponding to the head of the table.
Response: We aimed to provide the maximum information to the readers to make the tables standalone for readers. We have now modified the text in Tables 1,2,3 (now 2,3,4) to simplify and make it easy to understand. Reviewer 2 has appreciated the organisation of the first 3 tables (please see comment No. 4 below). We have also rearranged Table 4 (now Table 5); Table 5 is organised alphabetically based on the parasite’s names.
Minor revisions:
- L41-42-43: TMD instead of TBDs (Lines 41,43,44)
Response: TBDs are changed to TMDs as suggested.
- L77: change the verb typified which is not adequate
Response: The verb typified is replaced by “such as”. The sentence now reads as “In addition, it has been suggested that anthelmintics such as ivermectin (IVM), levamisole (LEV) and thiabendazole (TBZ) are substrates of ABC transporters.” (Line 78)
- Figure 1: the quality of the figure should be improved and in particular the AD which is difficult to read.
Response: Figure 1 has been revised.
- L100-101: this sentence should be reformulated since it’s difficult to understand if helminths has or not the same number of abc transporter genes as mammals.
Response: The sentence has now been rephrased as “Helminths possess a larger number of ABC transporter genes than mammals, which have a few multidrug resistance (MDR) transporters”. (Lines 102-103).
- All in vivo and in vitro terms should be written in italics
Response: We have rechecked the manuscript and made sure that all in vivo and in vitro terms are written in italics
- Page 7: the efflux instead of ethe fflux
Response: Corrected. (Line 320)
- First sentence of paragraph 5.3: since…, they may regulate instead of these may
Response: Modified as suggested. (line 363).
Reviewer 2 Report
Please see the attached file.

The language is in high quality with a consistent British style.
Author Response
The authors thank the reviewers for their positive feedback and relevant comments.
Please note that the line numbers referred to in response to the reviewer's comments correspond to the line numbers in the tracked changes version of the manuscript.
Response to Reviewer 2
In the Manuscript ID: pathogens-2382079, Abeer and colleagues reviewed the current research status on ABC transporters related to the survival of parasitic nematodes and discussed the prospect for the development of new control strategies using the ABC transporters as drug targets. Parasitic nematodes are important pathogens involved in both plant and human diseases. Nematodes have multiple and diverse ABC transporters, which play crucial roles in their survival. Therefore, ABC transporters, especially the Pgp protein, could be used as drug targets to develop novel control strategies for parasitic nematodes. The review covers the gene expression of ABC transporters in nematodes, the physiological roles of ABC transporters, research methods for ABC transporters, the involvement of ABC transporters in drug resistance, and the potential of ABC transporters as drug targets and provides recommendations for future research. This is a timely analysis of an important topic. Overall, the manuscript is well-written. The logic is very straightforward and simple to read and comprehend.
The reviewer did not find major issues but has some suggestions for minor revision.
- In this manuscript, the authors enumerated all 7 subtitles, including Introduction..., Conclusions and Recommendations. This is not necessary.
Response: We acknowledge the reviewer’s point of view; however, the numbering was done following the journal’s format and to minimise confusion for the readers as the paper has several sections and subsections,
- In the introduction, TBD was not defined. Is it a mistake for TMD?
Response: This has been fixed now (Lines 41,43,44).
- In the introduction, the authors introduced that eukaryotic ABC transporters can be divided into seven families and further stated that functional ABC systems can be divided into three major classes but didn’t discuss the relationships between the 7 families and the 3 major classes. This makes the classification of ABC transporters confusing. The reviewer suggests only introducing the 7 families or give a further introduction to the relationships between the 7 families and 3 classes.
Response: We acknowledge the reviewer’s concern here. We have rephrased the text. Based on genomic analysis, eukaryotic ABC transporters have been categorised into seven sub-families from A to G [2–4]. Two of the ABC transport protein families (ABCE and ABCF) lack TMDs, thus do not act as transporters, and are associated with other cellular processes, for example, ribonuclease inhibition and translational control [5,6]. ABC transporters have diverse structures depending on different domain compositions and ATP binding sites; thus, ABC transporters are classified as full transporter (complete structure with two NBDs and two TMDs), which contain two ATP-binding sites and the proteins with one ATP-binding site, the half transporters (half structure with one NBD and one TMD). Some transporters carry only a single NBD or TMD are termed single domain structures. In contrast, only NBDs are present at the N- and the C-terminus in non-transporter ABC proteins [7]. Based on the functional classification, ABC transporters are divided into three major classes; Class 1 and 2 are involved in the translocation of substrates across the cell membrane, and Class 3 is mainly associated with DNA repair and regulation of gene expression in cells. (Lines 34-43)
- Tables 1, 2 and 3 have been organized well alphabetically; however, Table 4 appeared to be disordered. The second “Inhibitors used Nematode spp. Study Design Effect on anthelmintic efficacy/availability Reference(s)” in the table is redundant.
Response: We have also rearranged Table 4 (now Table 5). Table 5 is now organised alphabetically based on the parasite’s names.
- Figure 1 shows that ABC transporters across two cell membrane bilayers. This looks not right. The TMD of ABC transporters is usually only located on a lipid membrane bilayer but not across two cell membrane layers.
Response: Thank you for pointing this out. We have now corrected Figures 1 and 2.
- References are in a consistent style and well organized. The last reference has a minor problem, 150. before the authors is redundant, please remove it.
Response: This has now been corrected.
Round 2
Reviewer 1 Report
The second version has been greatly improved with more readable figures and new tables allowing synthesis of information. Nevertheless, there are still problems with the tables.
Table 1: 2 lines lack the word genes at the end (F. gigantia & T. canis)
Table 2: sentences with verb should be avoided and reformulated
Table 3 : drug should be abbreviated to gain place for other columns. For H. contortus, it lacks the drug (there is only the concentration written).
Table 4 : it lacks the name of the drug used on the first line. Informations in sensible and resistant isolated should be separated in columns (versus line for the moment) to distinguish both isolate results.
Table 5 : if the priority is given to the name of the nematode, this information/column should be placed on the left, within the first column, followed by a column specifying the name of the drug. Abbreviations should be used as for Table 3. Lines of the tables should be uniform in terms of thickness
Author Response
The second version has been greatly improved with more readable figures and new tables allowing synthesis of information. Nevertheless, there are still problems with the tables.
We thank the reviewer for their time reviewing the manuscript and making constructive comments.
Table 1: 2 lines lack the word genes at the end (F. gigantia & T. canis)
Response: This has now been fixed by adding the word "gene" as suggested.
Table 2: sentences with verb should be avoided and reformulated
Response: This has now been fixed, as suggested by modifying the sentences with verbs.
Table 3: drug should be abbreviated to gain place for other columns. For H. contortus, it lacks the drug (there is only the concentration written).
Response: Drug names have been abbreviated as suggested. For H. contrortus, the name of the drug is now added.
Table 4 : it lacks the name of the drug used on the first line. Informations in sensible and resistant isolated should be separated in columns (versus line for the moment) to distinguish both isolate results.
Response: This has now been fixed. The drug name is added on the first line. The cell with resistant and susceptible isolates is now read as resistant vs susceptible isolate.
Table 5 : if the priority is given to the name of the nematode, this information/column should be placed on the left, within the first column, followed by a column specifying the name of the drug. Abbreviations should be used as for Table 3. Lines of the tables should be uniform in terms of thickness.
Response: As suggested, the table is now modified by moving the Nematode name column to the first place. Abbreviations have been used where possible. The line thickness in the table is 1.0 for all rows.
Round 3
Reviewer 1 Report
In the last manuscript submitted, the table 4 has a big problem and the word document only contain and display the first column. could you change and revise this and submit a new version ?
Author Response
In the last manuscript submitted, the table 4 has a big problem and the word document only contain and display the first column. could you change and revise this and submit a new version ?
Response: Thanks for your time. The appearance of the table was like this in the track changes version, however, the table is fine in the clear version. We have uploaded the revised version of the manuscript to verify that there are no formatting issues.
Round 4
Reviewer 1 Report
The table 2 should be placed closed to the end of the paragraph 5.2 to be into an entire form.
This will help to place the other tables 3 and 4 to be placed at the right place corresponding to their citation. In table 5, inhibitors should be abbreviated to gain place.
On the first line of figure 2, place the word helminth at the line.